# Viruses of Yams (*Dioscorea* spp.): Current Gaps in Knowledge and Future Research Directions to Improve Disease Management

**DOI:** 10.3390/v14091884

**Published:** 2022-08-26

**Authors:** Mame Boucar Diouf, Ruth Festus, Gonçalo Silva, Sébastien Guyader, Marie Umber, Susan Seal, Pierre Yves Teycheney

**Affiliations:** 1INRAE, UR ASTRO, F-97170 Petit-Bourg, France; 2CIRAD, UMR AGAP Institut, F-97130 Capesterre-Belle-Eau, France; 3UMR AGAP Institut, University Montpellier, CIRAD, INRAE, Institut Agro, F-97130 Capesterre-Belle-Eau, France; 4Natural Resources Institute, University of Greenwich, Central Avenue, Chatham Maritime, Kent ME4 4TB, UK; 5CIRAD, UMR PVBMT, F-97410 Saint Pierre, France; 6UMR PVBMT, Université de la Réunion, F-97410 Saint-Pierre, France

**Keywords:** Yam, virus, diversity, diagnostic, impact, epidemiology, control

## Abstract

Viruses are a major constraint for yam production worldwide. They hamper the conservation, movement, and exchange of yam germplasm and are a threat to food security in tropical and subtropical areas of Africa and the Pacific where yam is a staple food and a source of income. However, the biology and impact of yam viruses remains largely unknown. This review summarizes current knowledge on yam viruses and emphasizes gaps that exist in the knowledge of the biology of these viruses, their diagnosis, and their impact on production. It provides essential information to inform the implementation of more effective virus control strategies.

## 1. Introduction

Yam is a vernacular name used for various species of the genus *Dioscorea* (family *Dioscoreaceae*), which includes more than 600 species worldwide [1]. Some of these species are valued in tropical and subtropical regions where yams play a crucial role in food security and/or have a strong socio-economic importance [1,2,3]. Other yam species such as *D. pentaphylla*, *D. hamiltonii*, *D. oppositifolia*, and *D. polystchya* have interesting medicinal properties [1,4]. Approximately 95% of the world’s yam production originates from West Africa [2,5,6]. The main cultivation mode is based on vegetative propagation using yam tubers from the previous harvest. This, however, favors the accumulation and propagation of viruses due to the absence of sexual reproduction, which provides potent natural virus sanitation in plants since most plant viruses are not seed transmitted [7]. The control of viruses in yams therefore relies primarily on the use of virus-free planting material [5,8].

Virus infections have been reported in all yam growing areas, although geographical distributions vary between yam viruses (Table 1). The phytosanitary constraints caused by yam viruses on the production of clean seeds and the exchange of germplasm have prompted increased efforts to characterize the virome of yams and assist in the development of accurate diagnostic tools. 

Between 2012 and 2022, 14 new viruses have been found to infect yams [5,11,17,21,24,25,28,29,30,35,54]. These findings relied mostly on the implementation of high-throughput sequencing (HTS) technologies and associated bioinformatics pipelines. Generated sequence data enabled the development of robust molecular diagnostic tools for the detection of these newly characterized viruses, which in turn has assisted in the removal of infected planting material during the production of clean seeds through sanitation programs [8,42,43,56,57,58]. However, the biology, symptomatology, ecology, and impact on production of the majority of yam viruses remain unknown. These gaps in knowledge hinder the implementation of effective disease management strategies aimed at reducing the burden of viral diseases on production and the implementation of guidelines for the safer international exchange of yam germplasm.

This review addresses these gaps to help identify where additional efforts are needed for the design and implementation of effective control strategies, including more comprehensive sanitation programs. It is also intended to inform decision-makers with regard to the international exchange of yam germplasm.

## 2. Diversity of Yam Viruses

The International Committee for the Taxonomy of Viruses (ICTV) currently recognizes 25 viruses that infect yams, in 12 genera (*Aureusvirus*, *Ampelovirus*, *Badnavirus*, *Carlavirus*, *Cucumovirus*, *Dioscovirus*, *Fabavirus*, *Macluravirus*, *Potexvirus*, *Potyvirus*, *Sadwavirus*, *Velarivirus*) and eight families (*Alphaflexiviridae*, *Betaflexiviridae*, *Bromoviridae*, *Caulimoviridae*, *Closteroviridae*, *Potyviridae*, *Secoviridae*, *Tombusviridae*) (Table 1). Co-infection of viruses in yams appears to be common [5,8,11,12,28,29]. Knowledge of the diversity of viruses is key to improving the characterization of virus isolates, including their virulence, host range, or geographical distribution, which may shape epidemics. However, an important limitation in accurate diversity studies is the difficulty in getting access to diverse yam samples (species/varieties) from different geographical regions. This can be solved either through large scale samplings spanning entire regions or countries or through access to germplasm collections covering the genetic diversity and geographical distribution of *Dioscorea* spp., as exemplified in some diversity studies carried out on several yam viruses [11,17,25,28,30,55]. For example, Umber et al. [55] managed to characterize the molecular variability of *Dioscorea mosaic associated virus* (DMaV) RNA1 and RNA2 using a relatively low number of yam samples originating from different varieties and geographical areas conserved during a germplasm collection in Guadeloupe and Cote d’Ivoire. This approach enabled authors to demonstrate that both the host range and geographic distribution of DMaV were larger than initially thought. It also revealed that DMaV displays a high nucleotide variability in both its RNA1 (18.3%) and RNA2 (16.6%) and that intra-plant nucleotide variability was similarly very high in both genomic RNAs (17.7% for RNA1 and 16.6% for RNA2, respectively). Authors hypothesized that high genetic diversity could help DMaV escape sequence-specific host defense mechanisms.

Accessing yam samples from diverse geographical origins will also assist in studying the extent to which the diversity of specific yam viruses is determined by their geographical origin. Recent studies by Mendoza et al. [45] built on the efforts undertaken previously by Bousalem et al. [39] to unravel the geographical clustering of YMV diversity. Besides identifying a new phylogroup of YMV in Brazil, phylogeographic analysis [45] traced the most recent common ancestor of YMV in Cameroon back to around 1850 in plants of the *D. cayenensis-rotundata* complex. The spread of YMV to *D. trifida* in French Guiana was estimated to have occurred about 80 years ago, and more recently to *D. alata* in Burkina Faso and Puerto Rico about 53 and 30 years ago, respectively. Interestingly, Mendoza et al. [45] reported that the genetic diversity of YMV was more closely linked to geographical location rather than host yam species. A similar study carried out on YMMV supported an Asian–Pacific origin of YMMV in the *D. alata* species, followed by a geographical dispersion of this virus probably through the long-distance movement of infected germplasm [46]. Likewise, phylogeographic analyses carried out on CoV1, badnaviruses, YaV1, and DMaV shed light on the circulation, long-distance movement, plant-to-plant transmission modes, and host swaps of viral strains [17,25,28,30,55].

Mixed infections of multiple virus species and strains in vegetatively propagated crops such as yams provide conditions that favor the emergence of new strains via recombination [59]. Phylogenetic analyses showed that the emergence of new recombinant lineages played a key role in the evolution and spread of yam potyviruses [39,46,47,60]. Likewise, Bömer et al. [21] showed evidence for the recombination in badnaviruses between DBSNV as major parent and DBALV as minor parent in a DBRTV3 sequence. However, the role of recombination in the evolutionary process of yam viruses remains poorly documented overall and should be investigated further. Overall, phylogeography approaches have proved an efficient method for unravelling the complex processes driving the emergence, spread, and evolution of yam viruses [21,39,46,47,48]. It would be advisable to extend these studies to all viruses infecting yams.

## 3. Diagnosis

Virus diagnosis is key to efficient disease management, in particular during sanitation programs, which are the backbone of any clean seed production program. Inaccurate diagnosis leads to either false negatives or false positives that can have minor (e.g., misrepresentation of the distribution of viruses) to major impacts (e.g., the spread of viruses to new regions through the exchange of infected germplasm). For example, based on electron microscope observations, CMV was reported on *D. trifida* in Guadeloupe in 1977 [14]. Still, CMV presence has never been confirmed, despite repeated attempts based on large scale samplings and the use of sensitive RT-PCR-based CMV-specific diagnostics [8]. Additionally, CMV was reported to infect yam in West Africa at low incidences [15,16], but the analysis of high throughput sequencing datasets from *D. alata* and *D. rotundata* genotypes collected in Nigeria has not revealed any contig mapping to CMV [5,58], which raises doubt on the occurrence of CMV in Guadeloupe and Nigeria. Likewise, the diagnosis of DLV based on observations of flexuous rod particles by electron microscopy [9,10] proved unreliable. Sensitivity thresholds are also an issue for detecting yam viruses present at low titers [44,58]. Low virus titers can lead to false virus-negative certifications of yam plants, which can favor the spread of viruses through the unintentional exchange of infected germplasm nationally or internationally and have disastrous consequences for crop production and food security. Major efforts have been, and are still, placed on the development of accurate, sensitive, and cost-effective tools for the diagnosis of yam viruses [8,38,41,42,43,58,61]. However, such development can be particularly challenging because most yam viruses display a high molecular variability that can affect the accuracy and sensitivity of serological and molecular diagnoses [40,48,62,63].

In the last four decades, a wide range of serology-based tests, including immuno-electron microscopy (IEM), dot blot immunoassay (DBIA) and direct tissue blot immunoassays (DTBI), immunosorbent electron microscopy (ISEM), and variations of the enzyme-linked immunosorbent assay (ELISA) have been developed for the detection of yam viruses [8,18,64,65,66]. ELISA remains widely used for routine diagnosis [15,26,45,67,68] because of its simplicity and cost efficiency when handling a large number of samples, but it has several limitations. Firstly, antisera production is time-consuming and expensive, therefore antisera are currently only commercially available for a limited number of yam viruses including YMV, YMMV, CMV, and BBWV2 [15,68,69]. Broad-range polyclonal antisera capable of capturing all virus isolates are scarce and commercially available only for potyviruses [8,49,70]. Secondly, antigenic diagnostics do not generally allow for the differentiation of closely related variants, as reviewed by Boonham et al. [71]. Therefore, serological diagnosis of yam viruses is now supplanted by molecular diagnostics, which rely primarily on PCR or isothermal amplification-based methods (Table 1).

The use of PCR for virus diagnostics has greatly improved the sensitivity, specificity, and speed of yam virus indexing compared to that achieved by serological tests [61]. PCR has allowed for further characterization and increased knowledge of the genetic diversity and evolution of yam viruses [32,33,40,46,50,61,63], which in turn has helped refine PCR-based diagnostics. However, inhibitory substances present in yam tissues can cause false-negative results, limiting the reliability of PCR for yam virus diagnostics [38,43]. Major efforts have been placed on the extraction of high-quality nucleic acids from different types of yam tissue to limit this problem [8].

Immunocapture-PCR (IC-PCR) is a sensitive hybrid assay that combines the general specificity of antibodies for trapping viral particles with the increased specificity and sensitivity of PCR amplification of viral DNA sequences [38]. This technique overcomes the challenge of PCR-inhibitors that are often co-extracted with yam nucleic acids and the low sensitivity and specificity of ELISA, which can result in false-negative results associated with low virus titers [38]. IC-PCR has been reported to be at least five times more sensitive than ELISA and PCR techniques when used for detecting yam potyviruses [38,51,65,66]. However, the relative scarcity of antibodies directed against yam viruses limits the range of viruses that can be detected by IC-PCR (Table 1).

Quantitative PCR (qPCR) is less commonly used for the routine diagnosis of yam viruses, although it was reported to be more sensitive than conventional RT-PCR for the detection of YMV [44]. Further studies are required to evaluate the specificity and sensitivity of qPCR in detecting yam viruses, especially within clonally propagated plants in sanitation programs when viruses are likely to be present at low titers [72].

The detection of badnaviruses in yams is hampered by the presence of endogenous Dioscorea bacilliform virus sequences (eDBVs) and the high genomic and serological variability of badnaviruses [19,26,62]. PCR-based diagnostics cannot be used with confidence because they have been shown to generate false positives in some species as a result of the amplification of eDBVs [19,62]. Serological and immuno-serological tests, including ELISA, IC-PCR, and ISEM, can generate false negatives because currently available antiserum is insufficiently specific to all yam badnaviruses, with the best mixed polyclonal antisera (‘BenL’) no longer being available [62]. False positives have also been a problem through cross reaction with filamentous viruses or through the trapping of yam plant DNA containing eDBVs [18,62], which can be resolved by treatment with DNase following the immunocapture step of IC-PCR [24].

PCR-denaturing gradient gel electrophoresis (PCR-DGGE) [57,73,74] has been proposed to overcome these limitations, but it is not well suited for routine diagnosis of large numbers of samples. Furthermore, DGGE is also limited by the finding that it is unlikely to detect any target DNA less than 1 % of the total target pool [75]. Hence, episomal DBVs may not be detected by DGGE because of their low titers [73], and further studies are required to explore the benefits of DGGE in yam badnavirus diversity studies and assess its sensitivity for detecting episomal DBVs.

Episomal DBVs are circular DNAs, whereas endogenous DBVs are linear DNAs [76]. Rolling circle amplification (RCA) has been used to differentiate between episomal and endogenous DBVs [25,57], allowing for the identification and characterization of complete genomic sequences of both characterized and previously undescribed DBVs [25]. However, RCA is faced with the challenge of off-target amplification of plant genome-derived DNAs, such as mitochondrial or chloroplast DNA [25,77], as reported in sweet potato [78] and sugar beet also [79]. Sukal et al. [57] addressed this issue by optimizing and comparing random-primed (RP)-RCA, directed (D)-RCA, and specific-primed (SP)-RCA combined with high throughput sequencing (HTS) for the detection of DBVs. The optimized SP-RCA and D-RCA methods produced a significantly greater percentage of reads (~85%) mapped to the target viral genome compared to the RP-RCA protocol (~1%) reported in their study and a previous study [80]. These results highlight the potential of SP-RCA or D-RCA combined with HTS as useful tools for diagnosing DBVs and the characterization of their genomes, although these methods are not suited for processing large numbers of samples and are not cost effective.

Isothermal amplification methods use a single temperature for DNA amplification and require only a simple, low-cost heat source rather than more expensive thermal cycling equipment [81]. To date, recombinase polymerase amplification (RPA) as well as loop-mediated amplification (LAMP) assays have been successfully developed for the detection of three yam potyviruses: YMV, YMMV, and JYMV [42,43,82]. LAMP and RPA are more rapid than ELISA and PCR techniques, and have been reported to detect YMV and YMMV from positive samples in less than 30 minutes [41,42,43]. Furthermore, RPA and LAMP techniques require a single temperature of 37 °C and 65 °C, respectively, which eliminates the need for sophisticated equipment and makes them suitable for in-field diagnostics and uses in low resource settings [41,43]. These assays have also been reported to be more tolerant than PCR to inhibitory substances present in yam nucleic acid samples [42,43]. RPA was shown to have the same level of sensitivity as conventional PCR in YMV detection [41], whereas LAMP was around 100 times more sensitive than conventional PCR [48]. A drawback of the LAMP assay is that it requires a set of six primers, which can be challenging to design, especially when dealing with genetically highly variable viruses such as YMMV and yam badnaviruses [26,46,48,63,83]. Despite the advantages of RPA and LAMP techniques in virus diagnostics, these are recently developed techniques with only a few reports on their use for the diagnosis of yam potyviruses to date. The development of RPA and LAMP assays for further yam viruses would be beneficial to enable comprehensive rapid-screening in sanitation programs, trans-border transfer of valuable germplasm, and in-field diagnostic applications.

While serological and classical molecular diagnostics are target-specific, HTS circumvents this limitation by generating sequences from both known and unknown pathogens, which removes the need for prior knowledge of target sequences [84,85]. The use of HTS has led to the discovery and molecular characterization of numerous new yam viruses in the genera *Ampelovirus*, *Badnavirus*, *Macluravirus*, *Potyvirus*, and *Sadwavirus* (Table 1; [5,28,29,36,54,58,86]). HTS has proved very versatile and well suited for filling knowledge gaps around the diversity of yam viruses, but it requires an informed choice of samples and a sequencing platform. Although the Illumina technology yields a high nucleotide accuracy of ∼99%, Filloux et al. [36] identified two major biases associated with its use for characterizing yam viruses. Firstly, de novo assembled reads are often chimaeras of reads from different variants [87]. Secondly, the inability to build large contigs, due to the short length of the Illumina reads, can reduce the reliability of assembled viral genomes, even when they are mapped to known viral genomes using alignment-based approaches [88]. Third-generation sequencing approaches, such as Oxford Nanopore MinION [36,89,90,91] or PacBio, generate longer reads that are better suited for de novo genome assembly applications, which can overcome the biases of short-read approaches [87,92,93]. Filloux et al. [36] established that MinION is efficient for reliably detecting poly(A) tailed positive-sense single-stranded RNA (+ssRNA) yam viruses and accurately sequencing their genomes. This makes HTS-based detection a major asset to yam virus diagnostics, especially in seed yam production systems and sanitation programs [58], for which sensitive virus indexing is critical to assess the success rate of virus elimination [8,44,94] and avoid the multiplication of infected plants. Given its high sensitivity threshold, HTS is well suited for the detection of viruses with low titers such as those observed during cell culture in clonally propagated crops such as yams [58]. Although HTS workflows have been developed for yam detection [36,57,58], they have not yet been incorporated into yam sanitation programs or seed systems, likely because the high cost associated with HTS remains a major limitation. Multiplexing more than one sample on the same run could make HTS more cost-effective, but additional work is required to assess to what extent this would affect sensitivity and suitability for testing in vitro plantlets, especially for viruses with low titer. Another limitation of HTS is that the bioinformatics analysis of HTS datasets requires expertise and computer and data storage resources that may not be readily available [93]. Additional work is needed to develop bioinformatic pipelines suitable for operators with limited bioinformatic expertise that can be used to analyze HTS data.

The detection of viral sequences by HTS without any biological information creates additional questions and challenges for decision making. Can valuable germplasm containing these viral sequences be exchanged? What threat does the virus pose to plant health? In the absence of biological information on these viruses, reliable decisions on the risk of seed exchange cannot be made. Plants with newly discovered viruses can only be exchanged where they can be grown under strict restrictions to avoid any possibility of virus spread. Before release, further studies would need to be carried out on the symptomatology and impact of the viruses, alone and in mixed infections, as well as surveys to determine the distribution of the newly identified viruses in existing germplasm and the territory of introduction to assist the decision-making process.

## 4. Symptomatology and Impact on Production

Symptoms commonly associated with virus infections (mosaic, chlorosis, necrosis, mottling, leaf deformation or distortion) are often observed on infected yam leaves (Figure 1). However, co-infections make it difficult to elucidate the etiology of yam viral diseases, the impact of individual viruses and/or synergistic infections on plant growth and yields [11,29], and the possible beneficial role of some viruses on yams. In fact, the biology of most yam viruses remains largely unknown, as is often the case for viruses newly identified by HTS approaches for which biological characterization and risk assessment studies have not yet been performed [95].

Infectious virus clones have helped unravel virus–host and virus–virus interactions for numerous plant viruses [96,97,98], including their symptomatology in single or mixed infections [99]. An infectious clone is available for only one virus infecting yams, CYNMV. Using this infectious clone, Kondo et al. [33] showed that CYNMV infection is associated with systemic necrotic mosaic symptoms in the yam variety ‘Nagaimo’. A more accessible approach relies on the use of partially sanitized plants bearing various combinations of single or co-infections to try to unravel the symptomatology of yam viruses [8].

Assessing losses due to plant pathogens is a difficult task due to the number of parameters that must be considered to draw valid statistical analyses [100,101]. Very little data is available for losses attributable to yam viruses, apart from small-scale studies carried out on YMV, JYMV, and CYNMV infections for which yield losses were recorded and quantified [33,50,60,102]. Thouvenel et al. [102] and Egesi et al. [103] attempted to assess the impact of YMV on yam production. However, they pointed out that factors such as co-infections with other viruses, environmental, or biotic parameters that may influence virus loads and symptom severity were not all considered in their study, which makes it difficult to accurately assess the proportion of the loss attributable to YMV infection. Adeniji et al. [104] designed a more suitable, although small-scale, experimental system in Nigeria, in which they manually inoculated YMV on *D. rotundata* plants and assessed the impact of infection on a range of physiological traits and yield. For this, they performed virus indexing targeting YMV, YMMV, DaBV (BDALV), and CMV using TAS-ELISA, whose sensitivity and specificity is not optimal, as discussed above. Symptoms and impact of YMV infection on plant growth and yield losses were registered and compared between infected and mock-inoculated plants. However, the presence and impacts of other viruses (such as YVY, YaV1, or DMaV) that have since been identified and shown to be widespread in West Africa [5,17,23,24], were not monitored and may well have impacted results obtained in the experiments of Adeniji et al. [104]. HTS of yam samples used in future experiments is advised to make interpretation of biological data more robust.

For a better estimation of the impact of viruses on yam production, the sole evaluation of yield losses based on statistical comparisons between yields of infected and healthy plants is not sufficient. Several other criteria are impacted by viral infections and must be considered. For example, decreased tuber size and quality reduces the seed pool available for the next planting and makes harvesting more difficult and time consuming, incurring additional labor costs. Likewise, decreased tuber food palatability negatively impacts consumer preferences and marketing channels. In addition, estimation of yield losses should be based on a comprehensive approach using a conceptual framework involving virologists, statisticians, and economists, similar to what has been achieved for other vegetatively propagated crops [100,105,106,107,108].

In this regard, the use of severity scores based on leaf symptoms as a proxy of infection status or resistance to viruses [68,94,109], although indicative of plant health in some situations, should be avoided. Such severity scores can be biased by co-infections by several viruses, which sometimes results in synergistic symptoms, the presence of other pathogens, the lack of knowledge on virus symptomatology, or the confusion between viral symptoms and nutritional deficiencies. Moreover, inconsistencies between such severity scores and the presence of viruses were reported in yams [12,109]. Other methods using sensors and artificial intelligence to measure viral symptoms, such as near-infrared spectroscopy (NIRS), could provide a reliable alternative to the visual observation of symptoms [110], although they require sophisticated equipment.

## 5. Epidemiology: Transmission, Reservoir Plants and Cultural Practices

Vegetative propagation plays a predominant role in the persistence and spread of viruses in many important crops [7]. In yams, it is hypothesized that the use of infected cuttings or infected tubers as planting materials is the main driver of virus transmission [6]. Several studies have shown that the secondary or horizontal spread of viruses infecting vegetatively propagated crops involves insect vectors and/or mechanical transmission [37,50,64,102,104,109,111,112]. In yams, non-persistent transmission by aphids has been reported for BBWV2, CYNMV, JYMV, YMV, and YMMV [37,50,53,64,102,109,111] as well as transmission of DBALV by the mealybug species *Planococcus citri* [23], whereas mechanical transmission has been reported for YSV, YLV, YMV, and YMMV [13,56,102,104,109,111]. However, the transmission mode of other viruses infecting yams remains unknown [5,11,17,20,21,24,25,28,29,30,34,35,54], which prevents the development and implementation of control strategies against these viruses.

The possible role of weeds as reservoirs and the impact of farming practices on the epidemiology of yam viruses has not been extensively addressed either [113], although these factors could be significant levers for controlling these viruses through better weed management and/or the use of clean seeds [114,115]. In fact, the sustainable control of yam viruses would be best achieved by combining several approaches, such as frequent seed renewal (using virus-free seed), weed control (mulching, weeding), and/or control of vector populations.

Very little is currently known about how yam viral diseases are spread through the informal trade or exchange of infected plant material. Social epidemiology approaches [115] could prove useful for identifying the cultural factors that govern social interactions between farmers, including exchanges of yam germplasm. It is worth noting that community-based approaches for managing germplasm exchange between farmers can increase the efficiency of disease control programs based on the use of clean seeds [114]. Likewise, innovative tools, such as ‘Seed Tracker’, allow for the electronic certification of seeds [116]. These approaches are mainly used for cassava, but they could be adapted to any root and tuber crop. Defining the socio-cultural systems and determinants that structure the exchange or purchase of seeds and agricultural practices could help identify the economic and social constraints that influence the deployment of control strategies. An example is the use of yam clean seeds, which has encountered difficulties in West Africa through its adoption by farmers because it implies changes from traditional practices such as using tubers from the previous harvest as planting material [117].

## 6. Discussion

This review complements the efforts undertaken in recent years by the scientific community and stakeholders to mitigate the impact of viral diseases on yam production [118]. Particular emphasis is placed on the gaps that exist in the knowledge of the biology of these viruses, their diagnosis, and their impact on production because filling these gaps is critical to inform decisions that enable more effective control of the most damaging yam viruseses [119,120].

Vegetative propagation promotes the accumulation and spread of viruses. It results in the presence of diverse variants in plants and/or in co-infections by distinct viruses. In yams, this situation challenges accurate viral diagnostics. The ability of molecular techniques such as PCR, HTS, and isothermal amplification to overcome this problem makes them particularly appealing. HTS can also inform the host range and epidemiology of yam viruses—in particular, the role of reservoir plants such as weeds—in their spread and help unravel antiviral defense mechanisms through the profiling of yam virus-specific small interfering RNAs, as has been achieved for other crops [121,122,123]. Efforts should be placed on optimizing HTS-based approaches and making them more cost-effective. In addition to HTS, which facilitates the identification of distinct viruses in mixed infections, several diagnostic tools are also available to assist in the characterization of mixed infections. For example, in the case of DMaV, multiple isolates can be detected [55], which is relevant for diversity studies. However, many of the other molecular tests detect yam viruses of the same genus but do not discriminate them at the species level [8,24,57,74]. Cloning followed by sequencing or the use of a multiplex PCR test based on specific primers is then needed for species identification and diversity studies.

Most yam viruses have an international distribution (Table 1), in part presumably due to past exchange of infected planting material. Many were discovered recently (e.g., DMaV, YaV1, CoV1) and, as a result, have not been regulated quarantine pests in the past. It is likely that the geographical distribution of these viruses is underestimated currently because the exchange of yam germplasm, which has been standard practice for a very long time, is likely to have promoted their spread at times when a diagnostic was not available for their detection. Thus, research efforts aimed at an improved knowledge of the distribution and impact of these viruses ought to be undertaken at international scale.

Elucidation on the epidemiology of yam viruses is critical for developing comprehensive control strategies and relies primarily on the characterization of biological vectors such as insects because members of all viral genera reported in yam, except the genus *Aureusvirus*, are known to be transmitted by insects. However, vector competence studies are complex to perform, time consuming, and provide a lower scientific “return on investment” than most other fields of virology, hence why the current wide gap in knowledge on vector-borne transmission of yam viruses, which also applies to the role of weeds in the epidemiology of yam viruses and to the biology of these viruses for similar reasons, including their symptomatology. In potato, vertical transmission is known to be only partial for potato virus X, Andean potato mottle virus, potato virus Y, and potato leafroll virus, which means that only a proportion of daughter tubers are infected in turn [124,125]. Whether a similar situation exists in yams is still unknown and should be investigated for all yam-infecting viruses.

Yams host a wide range of endogenous viral elements (EVEs) originating from badnaviruses [22,62,77] and geminiviruses [31,126], whose study helped refine the taxonomy of both genera [18,19,22,62] and improve molecular diagnostic tools [8,62,74,77]. However, the potential for infectious status EVEs in yams is not widely supported [22,62] and remains unknown, although vertical transmission of yam viruses through the activation of infectious EVEs would have major consequences on yam germplasm conservation and exchange as well as yam breeding strategies, as is the case for bananas [127]. Additional efforts are needed to elucidate the infectious or non-infectious status of yam EVEs using a combination of -omics, laboratory, and field approaches similar to that used successfully to unravel the role of infectious endogenous banana streak viruses in the epidemiology of cognate viruses [128,129]. Conversely, it has been hypothesized that some plant EVEs could be involved in resistance against cognate viruses [130,131], including in yams [24]. Lastly, badnavirus EVEs could be used as molecular markers to inform the evolutionary history of *Dioscorea* spp., in particular to distinguish *D. cayenensis* and *D. rotundata* species, in a similar manner to studies on eggplant [132].

The production of clean (virus-free) yam seed is based on sanitization by thermotherapy and meristem culture [8,58,133]. The effectiveness of some of these techniques varies according to the virus and yam species, some being easier to sanitize than others [8,134]. Sanitation strategies are, however, costly, time-consuming, and require both skilled personnel and specialized equipment. They cannot be widely implemented but large-scale production of virus-free plantlets is possible through small-scale plots where virus-free planting material could be multiplied under high sanitary standards including regular inspections and testing to identify and remove any infected plant. Sanitation programs face the additional challenge of deciding which viruses are the most critical for their virus indexing and which ones appear of no economic value. A way forward for the management of yam viruses is now to ensure that reports of novel viruses in yam are matched with research efforts to evaluate their biological properties, geographical distribution, prevalence, and impacts. Such information is essential to allow informed decisions on which yam viruses pose the greatest threat to yam improvement, breeding programs, and in longer-term food security.

Research efforts on the epidemiology, symptomatology, impact, and transmission of yam viruses have been scarce, probably due to the fact that yams remain understudied and underfunded compared to many other tuber crops [58]. For example, a search on the Web of Science with the keywords “Yam AND *Dioscorea*” and “Potato AND *Solanum*” for all publication categories revealed an average of 48 and 258 articles per year, respectively, during the last 20-year period (August 2002 to August 2022), reflecting a strong disparity in the research efforts made on these staple food crops and their pathogens. In the case of potato, the impact of certain viruses on yield has been successfully measured [119,120], several disease control strategies have been developed [114,119], the impact of seed exchanges on virus spread [120], and the interactions between viruses during mixed infections have been elucidated [59]. Similar advances have yet to be made for yams. Still, the nutritional and medicinal benefits, as well as the socio-economic significance of yams to millions of growers and consumers worldwide, advocate the need for enhanced research efforts in order to address the challenges outlined in this review.

## Figures and Tables

**Figure 1 viruses-14-01884-f001:**
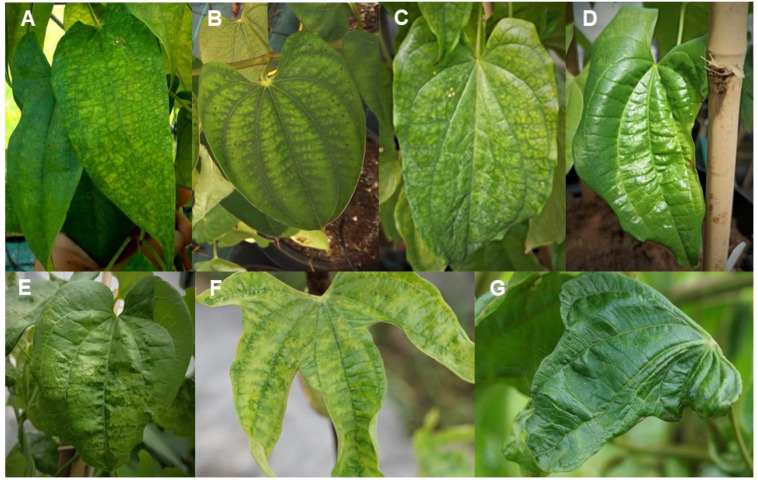
Examples of some viral symptoms observed on yam leaves. (**A**) Mosaic and leaf deformation on *D. alata* ‘sweet yam’; (**B**) Chlorosis on *D. alata* ‘Roujol’; (**C**) Mosaic on *D. rotundata* ‘Pouna’; (**D**) Mottling and leaf distortion on *D. rotundata* ‘TDr 07/00033′; (**E**) Puckering on *D. rotundata* ‘TDr 07/00033′; (**F**) Mosaic on *D. trifida* ‘Amoumbé 1′; (**G**) Severe leaf deformation on *D. trifida* ‘Praïnia’. Photographs courtesy of INRAE (**A**,**B**,**F**,**G**) and NRI (**C**–**E**).

**Table 1 viruses-14-01884-t001:** Viruses infecting yams.

Taxonomy				Genome	Genome Sequences *	Reported Yam Host Species **	Geographical Distribution	Available Diagnostic	Associated Symptoms	References
Family	Genus	Species	Recognized by the ICTV							
*Alphaflexiviridae*	*Potexvirus*	Dioscorea latent virus (DLV)	No	Monopartite, linear ssRNA(+)	No	Dco, Df	Puerto Rico	NA	Unknown	[9,10]
		*Yam virus X* (YVX)	Yes		**KJ711908**	Dr, Dt	Guadeloupe	RT-PCR	Unknown	[11]
		Yam potexvirus 1	No		KJ815100 KJ815099 KJ815098 KJ815097	Db, Dr, Dt	Guadeloupe, Haiti	RT-PCR	Unknown	[11]
		Yam potexvirus 2	No		KJ815103 KJ815102 KJ815101	Dn	Vanuatu	RT-PCR	Unknown	[11]
		Yam potexvirus 3	No		MN477413– MN477423	Dr	Côte d’Ivoire	RT-PCR	Unknown	[12]
*Betaflexiviridae*	*Carlavirus*	*Yam latent virus* (YLV)	Yes	Monopartite, linear ssRNA(+)	**KJ789130**	Do	China	NA	Unknown	[13]; Zou et al. (unpublished)
	Unassigned	Yam virus Y (YVY)	No		**MK782911** **MK782910**	Dr, Da, Dc	Cote d’Ivoire, Ghana, Guadeloupe, Nigeria	RT-PCR	Unknown	[5]
*Bromoviridae*	*Cucumovirus*	*Cucumber mosaic virus* (CMV)	Yes	Segmented, tripartite linear ssRNA(+)	EU274471 KX840389 KY766951 KY766950 LC191988	Da, Dr, Dt	Benin, Ghana, South Korea, Togo	ELISA, RT-PCR	Mottling, green veinbanding co-infection with YMV	[14,15,16]; Lee et al. (unpublished)
*Caulimoviridae*	*Badnavirus*	*Dioscorea bacilliform ES virus* (DBESV) DBV1 ^†^ or DeBV-A ^††^ **episomal**	Yes	Monopartite, open circular, double stranded DNA	**KY827394** AM072660 AM072663 AM072676 AM072700 AM072702 AM072700 AM072681	De, Da	Fiji, Papua New Guinea, Solomon Islands, Tonga, Vanuatu	RCA, IC-PCR	Unknown	[17,18,19]
		DBV2 ^†^ or DeBV-C ^††^ **unknown**	No		AM072688 AM072697 AM072679 AM072682	De	Papua New Guinea, Philippines, Vanuatu	RCA, IC-PCR	Unknown	[18,19]
		*Dioscorea bacilliform AL virus 2* (DBALV2) DBV3 ^†^ or DeBV-B ^††^ **episomal**	Yes		**MH404164** AM072674 AM072690 AM072704 AM072683 AM072685	Da	Papua New Guinea, Philippines, Solomon Islands	RCA, IC-PCR	Unknown	[17,18,19]
		*Dioscorea bacilliform SN virus* (DBSNV) DBV4 ^†^ or DsBV ^†^ **episomal**	Yes		**DQ822073**	Ds	Benin	RCA, IC-PCR	Unknown	[20]
		*Dioscorea bacilliform RT virus 3* (DBRTV3) DBV5 ^†^ or DBV-C ^††^ **episomal and endogenous**	Yes		**MF476845** **MG711312** **MG711311** EF466087 AM503358 AM503398 AM072659	Dr, Dt, Dpra	Benin, Fiji, French Guiana, Martinique, Nigeria	RCA, IC-PCR	Unknown	[18,19,21,22]
		DBV6 ^†^ or DeBV-D ^††^ **unknown**	No		AM072661 AM072680 AM072687	De	Fiji, Papua New Guinea	RCA, IC-PCR	Unknown	[18,19]
		DBV7 ^†^ or DeBV-E ^††^ **unknown**	No		AM072677 AM072689	De	Papua New Guinea, Philippines	RCA, IC-PCR	Unknown	[18,19]
		*Dioscorea bacilliform AL virus* (DBALV) DBV8 ^†^ or DBV-A(A)/DaBV ^††^ episomal and endogenous	Yes		**KX008571** **KX008572** **KX008573** **MH404174** **MH404172** **MH404167** **MH404166**	Da, Dr, Db, De, Dt, Dtra	Guadeloupe, Nigeria, Tonga, Vanuatu	RCA, IC-PCR	leaf distortion with veinal chlorosis symptoms, but infected plants can also be asymptomatic	[18,19,22,23]
		*Dioscorea bacilliform TR virus* (DBTRV) DBV9 ^†^ or DBV-B ^††^ **episomal and endogenous**	Yes		**KX430257** AM503374 AM503371 AM503377 AM503363 AM503369 AM503361 EF466065 AM072701 AM503359	Da, Dt, Dr, Ds, Dab, Dc, Dd	Benin, Burkina Faso, Cuba, Guadeloupe, Martinique, Republic of Guinea, Togo, Vanuatu	RCA, IC-PCR	Unknown	[18,19,22,24]
		DBV10 ^†^ or DpBV ^††^ **unknown**	No		AM072695	Dpen	Solomon Islands	RCA, IC-PCR	Unknown	[18,19]
		DBV11 ^†^ or DeBV-F ^††^ **unknown**	No		AM072662 AM072678 AM072686	De	Fiji, Papua New Guinea	RCA, IC-PCR	Unknown	[18,19]
		DBV12 ^†††^ or DBV-A(B) ^††^ **endogenous and presumably episomal**	No		KF830010 KF830003 KF830004 KF830005	Dr, Dc	Guadeloupe, Republic of Guinea	RCA, IC-PCR	Unknown	[18,19,22]
		*Dioscorea bacilliform RT virus 1* (DBRTV1) DBV13 **episomal**	Yes		**KX008574**	Dr	Nigeria	RCA, IC-PCR	Unknown	[25]
		*Dioscorea bacilliform RT virus 2* (DBRTV2) DBV14 **episomal**	Yes		**KX008577**	Dr	Nigeria	RCA, IC-PCR	Unknown	[25]
		DBV15 **presumably episomal**	No		AM944580 KX008584	Dr, Da	Cote d’Ivoire, Nigeria, Togo	RCA, IC-PCR	Unknown	[12,25,26]
	*Dioscovirus*	*Dioscorea nummularia associated virus* (DNUaV)	Yes		**MG944237**	Dn	Fiji, Samoa	RCA	Unknown—Potentially asymptomatic	[27]
*Closteroviridae*	*Ampelovirus*	*Yam asymptomatic virus 1* (YaV1)	Yes	Monopartite, linear ssRNA(+)	**MT409627**	Da, Dc, De, Dr, Dt	Guadeloupe, Nigeria, Vanuatu	RT-PCR	Unknown—Potentially asymptomatic	[28]
		*Air potato virus 1* (AiPoV1)	Yes		**MH206615**	De	USA	RT-PCR	Mosaic co-infection with DMV	[29]
	*Velarivirus*	*Cordyline virus 1 (CoV1)*	Yes		**OM471839**	Da, Dc, Dt	Guadeloupe, Vanuatu	RT-PCR	Unknown	[30]
*Geminiviridae*	*Begomovirus*	Yam yellow spot mosaic virus **presumably endogenous**	No	Circular, ssDNA	KJ854437	Do	People’s Republic of China	PCR	Unknown	Zhou et al. (unpublished)
		EGV1 and EGV2 **endogenous**	No		KJ629184– KJ629216 KJ629217– KJ629236	Do, Da, De, Df, Dn, Ds, Db, Dd, Dt	Burma, French Guiana, Haiti, Madagascar, Papua New Guinea, People’s Republic of China, south east Asia, Vanuatu, West Africa	PCR	Unknown	[31]
*Potyviridae*	*Macluravirus*	*Chinese yam necrotic mosaic virus* (CYNMV)	Yes	Monopartite, linear ssRNA(+)	**AB710145** **KU641566**	Do, Dof	Japan, People’s Republic of China, Republic of Korea	RT-PCR	Necrotic mosaic	[32,33]
		*Yam chlorotic mosaic virus* (YCMV)	Yes		**KT724961**	Dpa, Dz	People’s Republic of China	RT-PCR	Vein clearing, veinal necrosis, systemic chlorosis, mottling and mosaic	[34]
		*Yam chlorotic necrosis virus* (YCNV)	Yes		**MG755240** **MH341583**	Da, Dn	India, People’s Republic of China, Vanuatu	RT-PCR	Leaf yellowing, necrosis and mottling	[35,36]
	*Potyvirus*	*Yam mosaic virus* (YMV)	Yes		**U42596**	Da, Db, Dc, Dd, De, Du, Dj, Dn, Do, Dpo, Dr, Dt	Benin, Brazil, Burkina Faso, Cameroon, Colombia, Costa Rica, Cote d’Ivoire, Fiji, French Guiana, Ghana, Guadeloupe, Haiti, India, Japan, Madagascar, Martinique, Nigeria, Papua New Guinea, People’s Republic of China, Republic of Guinea, Republic of Korea, Reunion, Sri Lanka, Togo, Uganda, Vanuatu	Elisa, RT-PCR, CT-RT-LAMP, RT-RPA	Symptoms differ depending on yam varieties and include mosaic, vein banding, green spotting, flecking, curling and mottling.	[6,8,12,37,38,39,40,41,42,43,44,45]
		*Yam mild mosaic virus* (YMMV)	Yes		**JX470965**	Da, Db, Dc, Dd, De, Du, Dj, Dn, Do, Dpo, Dr, Dt	Benin, Brazil, China, Colombia, Costa Rica, Cote d’Ivoire, Fiji, French Guiana, Ghana, Guadeloupe, Guinea, Haiti, India, Japan Madagascar, Martinique, Nigeria, Papua- New Guinea, Reunion, South Korea, Sri Lanka, Togo, Uganda, Vanuatu	RT-PCR	Mild mosaic	[46,47,48,49]
		*Japanese yam mosaic virus* (JYMV)	Yes		**AB027007**	Dj, Dpo, Do, Da	China, Japan	RT-PCR, LAMP, IC-RT-PCR	Leaf mosaic and green-banding	[50,51]
		*Dioscorea mosaic virus* (DMV)	Yes		**MH206616**	Db	USA	RT-PCR	Mosaic during co-infection with AiPoV-1	[29]
		Yam potyvirus TGwadE2	No		AY821494	Dt	Guadeloupe	RT-PCR	Unknown	[52]
*Secoviridae*	*Fabavirus*	*Broad bean wilt virus 2* (BBWV2)	Yes	Segmented, bipartite linear ssRNA(+)	NA	Do, Dj	China, Japan, South Korea	ELISA, RT-PCR	Mosaic	[53]
	*Sadwavirus*	*Dioscorea mosaic-associated virus* (DMaV)	Yes		**KU215538 KU215539**	Da, Dc, De, Dr, Dt	Benin, Brazil, Guadeloupe, Nigeria, Cote d’Ivoire	RT-PCR	Mosaic	[54,55]
*Tombusviridae*	*Aureusvirus*	*Yam spherical virus* (YSV)	Yes	Monopartite, linear ssRNA(+)	**KF482072**	Dr	Nigeria	RT-PCR	Unknown	[56]

*: complete genome sequences are in bold. ** Da: D. alata, Db: D. bulbifera, Dc: D. cayenensis, Dco: D. composita, Dd: D. dumetorum, De: D. esculenta, Df: D. floribunda, Dj: D. japonica, Dn: D. nummularia, Do: D. opposita, Dpo: D. polystachya, Dr: D. rotundata, Ds: D. sansibarensis, Dt: D. trifida, Dtra: Dioscorea transversa, Dz: D. zingiberensis, Dpra: praehensilis, Dpen: D. pentaphylla. NA: not available. ^†^: according to Kenyon’s classification [18]. ^††^: according to Bousalem’s classification [19]. ^†††^: according to Umber’s classification [22].

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
