# Peer review of "Viruses of Yams (*Dioscorea* spp.): Current Gaps in Knowledge and Future Research Directions to Improve Disease Management"

_viruses, 2022, doi:10.3390/v14091884_

Round 1
Reviewer 1 Report
This is a very nice summary on the state of knowledge of yam viruses. The authors address the double-edged sword of HTS - we are now aware of so many viruses it is now critical to understand their biological significance.
Author Response
Reviewer #1:
We thank reviewer #1 for his compliments about our review.
Reviewer 2 Report
The manuscript number “Viruses-1860001” provides information and discussion about the current knowledge on yam viruses and future directions to manage the yam viral diseases. The authors address the diversity, the detection and diagnosis methods, the symptoms and disease effects, and the epidemiology of yam viruses. The manuscript is well prepared and will be useful to understand the current situation in the studies of yam viruses.
The following comments and suggestions need to be addressed:
1. The viruses listed in Table1 should be added with the information of their effects on yam production. Did they cause the viral diseases on yam in the field? Did they reduce the yam yield and quality? After all, not all virus infections could cause diseases.
2. Page 1. line 23:keywords “regulation” should be “control”.
3. Page 3. line 110: “D. trifida” should be italicized
4. Page 3. line 114: “D. alata, D. rotundata” should be italicized.
5. Page 4-13. The virus name should not be italicized. Please correct the entire manuscript and Table 1.
6. Page 15. line 225-226: The virus order, family, and genus should be in italic.
7. Page 17. Discussion: Authors should discuss the relationship between the diversity of yam viruses and their geographical distribution.
8. Page 18. line 404: “ D. cayenensis” and “ D. rotundata” should be italicized.
9. The diagnosis section should address the major yam viruses' symptoms and detection methods. The diagnosis of disease caused by a mixed infection should be discussed.
Author Response
Reviewer #2:
- The viruses listed in Table1 should be added with the information of their effects on yam production. Did they cause the viral diseases on yam in the field? Did they reduce the yam yield and quality? After all, not all virus infections could cause diseases.
These issues are addressed in part 4 of our review. Mixed infections by two or more distinct viruses, which are common in yams, make it impossible to correlate the presence of specific viruses with specific symptoms. Such a correlation could be established only for CYNMV, using an infectious clone. Likewise, the impact of a given virus on yield has never been quantified and is highlighted as an area needing attention in our suggested future research directions. Therefore, it is impossible currently to add accurate information regarding the symptomatology and impact of yam viruses in Table 1.
- Page 1. line 23:keywords “regulation” should be “control”.
Keyword has been modified accordingly.
- Page 3. line 110: “D. trifida” should be italicized
It was italicized in the original manuscript and modified by the online submission system.
- Page 3. line 114: “D. alata, D. rotundata” should be italicized.
Same comment as above.
- Page 4-13. The virus name should not be italicized. Please correct the entire manuscript and Table 1.
Only names of virus species recognized by the ICTV are italicized throughout the text and in Table 1.
- Page 15. line 225-226: The virus order, family, and genus should be in italic.
See answer to comment #3.
- Page 17. Discussion: Authors should discuss the relationship between the diversity of yam viruses and their geographical distribution.
This issue is discussed in the second paragraph of part 2 (lines 68-79).
- Page 18. line 404: “ D. cayenensis” and “ D. rotundata” should be italicized.
See answer to comment #3.
- The diagnosis section should address the major yam viruses' symptoms and detection methods. The diagnosis of disease caused by a mixed infection should be discussed.
See answer to comment #1 regarding symptoms. A figure illustrating virus symptoms has been added (see below).
A paragraph has been added in the discussion section to address the diagnosis of diseases caused by mixed infections (lines 289-294).
Reviewer 3 Report
This is a comprehensive review on plant viruses that infect yam plants. The manuscript is well written with sufficient information covering virological and epidemiological aspects of yam viruses. This review will be useful for understanding current problems and gaps in yam virus researches, and thus provide insights on future research direction for development of control strategies of virus diseases in yam. To make the manuscript more informative and appealing, my suggestions are, if possible, to add one figure showing yam plants with virus symptoms/diseases and one figure illustrating the geographical distribution of yam viruses (world map). I have minor corrections to the manuscript (pdf attached).

Author Response
Reviewer #3:
To make the manuscript more informative and appealing, my suggestions are, if possible, to add one figure showing yam plants with virus symptoms/diseases and one figure illustrating the geographical distribution of yam viruses (world map).
A figure (Figure 1) showing a range of typical virus symptoms has been added.
We purposely did not provide a figure illustrating the geographical distribution of yam viruses for two reasons : (i) we feel that it would be somehow redundant with the information provided in Table 1 and (ii) it could not fit on a single figure, considering that the geographical distribution of the 25 viruses cannot be disconnected from the geographical distribution of their hosts (18 yam species). I have minor corrections to the manuscript (pdf attached).
All corrections suggested by reviewer #3 were considered in the attached revised manuscript, with one exception. On line 193, reviewer #3 proposed to replace “low” by “high” but we think that we were right in our writing. Indeed, if the HTS methods do show a high detection sensitivity, it is because their sensitivity threshold is low.
Reviewer 4 Report
This nicely written mini-review provides gaps in virus-yam interplay and illustrates future directions in virus disease management. I only have a few suggestions, as below.
1, The authors point out the gaps in the virus-yam interplay, but the underlying reasons were not addressed. From the reader's perspective, it would be necessary to provide the challenges that raise the gaps or hinder the study of yam viruses.
2, It would be helpful to add a paragraph in the "Diversity of yam viruses" section to illustrate the taxonomy of the documented yam viruses. Though most of the information has been presented in Table 1, I suggest the authors summarize how many viruses can infect yam and which groups/families those viruses belong to.
3, Per the references, the order of many citations in the text may be wrong and need a revision. For instance, the number 54 was shown in line 43.
4, Typos and grammar mistakes. Line 128, "DBI" cannot stand for "dot", Here, a full name is needed. In addition, many acronyms may not be necessary—for example, PVX and PVY in line 387, which appeared only once in the text. Please revise the entire manuscript.
In Line 110, line 114, line 404, and many others, the species' scientific name should be italic.
Author Response
Reviewer #4
- The authors point out the gaps in the virus-yam interplay, but the underlying reasons were not addressed. From the reader's perspective, it would be necessary to provide the challenges that raise the gaps or hinder the study of yam viruses.
This point has been addressed at the end of the discussion section (lines 332--343).
- It would be helpful to add a paragraph in the "Diversity of yam viruses" section to illustrate the taxonomy of the documented yam viruses. Though most of the information has been presented in Table 1, I suggest the authors summarize how many viruses can infect yam and which groups/families those viruses belong to.
A paragraph summarizing the taxonomy of viruses infecting yams was added (lines 50-53).
- Per the references, the order of many citations in the text may be wrong and need a revision. For instance, the number 54 was shown in line 43.
The numbering of references has been thoroughly checked and revised where needed
Typos and grammar mistakes. Line 128, "DBI" cannot stand for "dot", Here, a full name is needed. In addition, many acronyms may not be necessary—for example, PVX and PVY in line 387, which appeared only once in the text. Please revise the entire manuscript.
Changes were made in the text accordingly.
- In Line 110, line 114, line 404, and many others, the species' scientific name should be italic.
See answer to comment #3 of referee #2.